# The Effects of Calcium, Magnesium, Phosphorus, Fluoride, and Lead on Bone Tissue

**DOI:** 10.3390/biom11040506

**Published:** 2021-03-28

**Authors:** Żaneta Ciosek, Karolina Kot, Danuta Kosik-Bogacka, Natalia Łanocha-Arendarczyk, Iwona Rotter

**Affiliations:** 1Department of Medical Rehabilitation and Clinical Physiotherapy, Pomeranian Medical University in Szczecin, Żołnierska 54, 71-210 Szczecin, Poland; ciosekzaneta@gmail.com (Ż.C.); iwrot@wp.pl (I.R.); 2Department of Biology and Medical Parasitology, Pomeranian Medical University in Szczecin, Powstańców Wielkopolskich 72, 70-111 Szczecin, Poland; nlanocha@pum.edu.pl; 3Independent Laboratory of Pharmaceutical Botany, Pomeranian Medical University in Szczecin, Powstańców Wielkopolskich 72, 70-111 Szczecin, Poland; kodan@pum.edu.pl

**Keywords:** bone, calcium, fluorine, lead, magnesium, phosphorus

## Abstract

Bones are metabolically active organs. Their reconstruction is crucial for the proper functioning of the skeletal system during bone growth and remodeling, fracture healing, and maintaining calcium–phosphorus homeostasis. The bone metabolism and tissue properties are influenced by trace elements that may act either indirectly through the regulation of macromineral metabolism, or directly by affecting osteoblast and osteoclast proliferation or activity, or through becoming part of the bone mineral matrix. This study analyzes the skeletal impact of macroelements (calcium, magnesium, phosphorus), microelements (fluorine), and heavy metals (lead), and discusses the concentration of each of these elements in the various bone tissues.

## 1. Introduction

Bones are calcified tissues composed of 5%−10% water, 50%−70% hydroxyapatite and 20%−40% organic components, including type I collagen and 10% noncollagenous proteins involved in bone mineralization [1]. Based on porosity and the unit microstructure, bone is classified into two types. Compact bone (also called cortical bone or dense bone) is dense bone tissue surrounding the medullary cavity, or bone marrow. Cancellous bone (also called trabecular bone or spongy bone) is a structure interspersed in the bone marrow. The adult human skeleton is composed of 80% compact and 20% cancellous bone, and the whole structure is highly vascularized [2].

Bone is continuously remodeled by alternating resorption and accretion. The balance between these two is tightly regulated by mechanical and hormonal stimuli to keep the bone structure. An imbalance leads to pathologies, such as osteoporosis, characterized by low bone density [3]. Bone growth and metabolism are also modulated by trace elements, such as iron (Fe), zinc (Zn), copper (Cu), calcium (Ca), phosphorus (P) and magnesium (Mg). It is assumed that both deficiency and excess of the trace elements may be a risk factor for the development of bone diseases such as osteoporosis [4,5].

Many elements have an impact on bone health through incorporation into the bone matrix, as well as the regulation of cellular resorptive and formative processes, including interaction with key enzymes [6]. Regarding the skeleton, elements have been classified into two groups: bone surface seekers and bone volume seekers. Bone volume seekers are diffused into the bone volume and placed on all bone surface types, with a preference for growing surfaces. They are also found in the calcifying areas of the skeletal cartilage [3]. The most common bone volume seekers are calcium, strontium (Sr), and lead (Pb) [7]. Their sorption mechanism can be seen as an ionic exchange at the mineral level. Ion exchange (mostly +2 cations, including Ca^2+^, Mg^2+^, Pb^2+^) occurs within the hydration shell, which surrounds the mineral crystals. It has been reported that this replacement is reversible, and is a likely precursor of exchange within the crystal matrix [8,9]. By contrast, bone surface seekers, also known as Pu-like elements, show no initial diffuse distribution into the bone volume. They are generally more concentrated on resorbing and resting bone surfaces [3]. The rate of uptake from blood to bone surfaces is the only variance in the adult pathway rate constants of Pu-like elements, including radionuclides [7].

The knowledge of the mechanisms of metal accumulation in bones, and the concentration of elements in bone tissue has not so far been fully understood [10]. It is assumed that the concentration and distribution of elements in the bone tissue are correlated with biological and environmental factors, including sex, age, diet, place of residence, smoking, alcohol consumption, and chronic diseases [11]. This review is focused on the impact of calcium, magnesium, phosphorus, fluorine (F), and lead on bone metabolism, highlighting the mechanisms of action that are most relevant to bone.

## 2. Calcium (Ca)

Calcium is a nutrient necessary for the proper functioning of the human body. This macroelement, which influences many extracellular and intracellular processes, is essential for the development, growth, and maintenance of bone, and for the stability of the cellular cytoskeleton [12]. Calcium regulates the activity of intracellular enzymes and participates in neuronal conduction through the ion channels. The total content of Ca in an adult human body is about 1200 g, which is about 2% of body weight [12]. About 99% of the body’s Ca is found in the bones and the teeth, stored as hydroxyapatite, which is responsible for the mineralization of tissues. The demand for Ca varies throughout life, depending on different growth needs in childhood and adolescence or during pregnancy and lactation [13].

Calcium intake determines skeletal Ca retention during bone growth, thus contributing to peak bone mass (PBM) achieved in early adulthood [14]. According to Zhu et al. [15], Ca intake was a less important yet a still significant predictor of the total body bone mass. Inadequate Ca intake during the growth period may have a negative impact on bone maturity, and thus predispose a person to an increased risk of osteoporotic fractures later in life. The analysis of white women included in the third American National Health and Nutrition Examination survey (*n* = 3251) revealed that those of them who consumed low amounts of milk in childhood and adolescence had low bone density in adulthood and a higher risk of fractures [16]. What is more, low bone mass is a potential contributor to childhood fractures. A report on New Zealand children demonstrated that those who avoided cow’s milk were at 1.7 times greater risk of fractures before puberty [17].

It has been found that increased dietary Ca intake improves bone mineral density (BMD) in a short time. According to Cashman and Flynn [18] the increased Ca supply during the growth period improves bone mineral accretion (BMA) by 1–5%, depending on the measurement site. The results of randomized studies confirmed a relationship between the amount of Ca consumed and bone mineral density. Bonjour et al. [19] reported an increase in BMD in eight-year-old girls who consumed an additional 850 mg of Ca daily, compared to girls on a normal diet. The greatest gain in BMD was observed in children whose Ca supply was previously the lowest. When examining children under the age of 18 without concomitant diseases or treatments affecting bone metabolism (n = 2859), Winzenberg et al. [20] noticed that Ca supplementation significantly improved the total body bone mineral content (BMC) and forearm BMD, but had no effect on lumbar spine or femoral neck BMD. According to Huncharek et al. [21], Ca or dairy intervention in participants with low Ca intake (450–750 mg/day) increased total body BMA. Based on their long-term study, Matkovic et al. [22] observed that supplementation with Ca had a significant positive effect on BMA in puberty; however, in late adolescence this effect was weaker. As stated by these authors, BMD can catch up during the bone consolidation phase to compensate for the impaired BMA during the growth spurt associated with insufficient Ca intake. Nevertheless, this study also made it clear that people with low calcium intake may not achieve complete catch-up and thus may not reach their target basic metabolic panel (PMB).

The deficiency of Ca is the main risk factor for osteoporosis. Changes in serum Ca levels involve adaptations in bone remodeling, as in the case of enhanced bone resorption caused by low serum Ca levels. Based on research on the effect of increased Ca intake on bone health in peri- and postmenopausal as well as elderly women, it was found that supplementation with Ca for more than five years reduces bone loss in postmenopausal women, especially those whose usual Ca intake is low (< 400 mg / day) [23,24].

The main factors that keep blood Ca on constant levels are 1,25- dihydroxyvitamin D3 (1,25(OH)_2_ D_3_), parathyroid hormone (PTH), and fibroblast growth factor 23 (FGF23) [25,26,27,28]. A constant concentration of Ca is essential for many biological functions, hence a change in serum Ca levels is the major inducer of Ca homeostasis. The inactivation of the calcium-sensing receptor (CaSR) in the parathyroid glands, caused by a decline in serum Ca, stimulates the release of PTH. PTH binds to its receptor in the kidneys, where it increases the reabsorption of Ca and the production of 1,25(OH)_2_ D_3_. Both circulating PTH and 1,25(OH)_2_ D_3_ bind their respective receptors on osteoblasts, thereby improving the expression of the receptor activator for nuclear factor κ B ligand (RANKL), which stimulates osteoclastic bone resorption and the release of Ca and P into the circulation. This restores Ca levels and triggers negative feedback mechanisms, including calcitonin release from the thyroid gland, which decreases renal Ca reabsorption and intestinal absorption, and inhibits osteoclastic bone resorption, thus keeping Ca levels within the optimal range [27,28].

The concentration of Ca in bones is determined by a variety of issues, among them sex, age, diet, physical activity, smoking, ethnicity, as well as genetic and endocrine factors [29]. In bones, the highest Ca levels are observed in people aged 20–39, and then the concentration of this element decreases. The elderly, and especially postmenopausal women show a reduced capacity to absorb Ca [30]. A decline in the absorption of Ca was also demonstrated in men and women over 60 years of age. It was found to be lower by one third in 70–79-year-old participants compared to those aged 20–59 years [30]. Analyzing the gender of participants, based on the data presented in Table 1, higher Ca levels were found in the bones of men than women. It is related to two common female hormones, estrogen and progesterone, which keep bones strong. The older a woman is, the lower level of hormones she has and the weaker her bones are [31]. Decreased Ca absorption also occurs in the course of many diseases, including malabsorption syndrome, primary cirrhosis, and celiac disease [32,33,34]. Calcium is lost from the bones in the event of complete physical inactivity, resulting in a rapid decrease in bone mass. This happens in people whose body parts are placed in a cast after injury. Similarly, the loss of Ca is observed in the case of individuals immobilized due to denervation, such as paraplegia and poliomyelitis, as well as astronauts in a gravity-free environment [32]. The loss of Ca from the bones during periods of inactivity is mainly caused by losing proteins from the bone matrix, which is associated with losing proteins from other parts of the body, especially the muscles. This calcium loss can also be induced by strenuous exercises. Moderate exercises improve bone mass, but very intensive effort may reduce it, even though local muscle activity increases local bone mass [32,33,34]. Diet has also a big impact on the Ca concentration in the bones. A positive correlation between the frequency of seafood consumption and the Ca content in the bones was reported [35]. Łanocha-Arendarczyk et al. [35] found that the consumption of game meat as part of the diet also influences the level of this macroelement in the bones. Higher Ca concentration was noted in the cortical bone of the patients with a game meat diet, while in the spongy bone, higher Ca levels were reported in the patients whose diet exclusive game meat.

The studies concerning Ca content in the various types of bone were studied in Asia, North America, and Europe (Table 1). The Ca level in the bone tissue can be arranged in descending order: north America>Europe>Asia. It is important to point out that we found only 1 paper from the USA, 6 articles from Asia, and more than 30 from Europe. The concentration of Ca between continents may vary due to the different diets of patients, but also due to the environmental pollution with heavy metals coming from industry. It has been reported that Ca concentration in the bone of patients from industrial areas is almost 10 times lower than in the bones of patients living in the areas without any huge factories [35,36].

The level of calcium varied depending on the type of human bones. Based on the available data presented in Table 1, the highest Ca concentration is found in the head of the femur, and the lowest in the femoral neck. This may be related to the fact that the femoral head is better vascularized and thus better nourished.

## 3. Magnesium (Mg)

As an element essential for the proper functioning of the body, Mg participates in many metabolic pathways in cells [55]. This element activates over 300 enzymes involved in the metabolism of carbohydrates, nucleic acids, and proteins [56]. It acts as a stabilizer of cell membranes, reducing their permeability. It is responsible for maintaining the homeostasis in the body by participating in the transport of electrolytes through cell membranes [57]. Magnesium ions are also necessary for maintaining the anatomical and functional integrity of various cellular organelles, including the mitochondria and ribosomes [58]. It was found that more than half of the amount of Mg in the cell nucleus is closely related to nucleic acids [59]. Magnesium stabilizes DNA molecules and also enables the maintenance of the tRNA tertiary structure. This element is responsible for the normal muscle contractility, neuron excitability, and the release of hormones and neurotransmitters [60]. Magnesium acts as an anticoagulant; it stabilizes the cell membranes of platelets and inhibits their aggregation and the formation of wall clots. It affects leukocytes by influencing the phagocytosis process and lymphocyte production, and reduces inflammatory reactions [61]. The human body weighing about 70 kg contains from 20 to 35 g of Mg [62]. Approximately 60%, 30%, 9%, and 1% of Mg is present in bones, muscles, soft tissues and intercellular fluids, respectively [63].

Magnesium is essential for bone development and mineralization. It stimulates the activity of osteoblasts and enzymes from the phosphatase group, which are involved in the bone formation process [64]. Magnesium deficiency directly affects bone density. Based on animal studies, it has been found that its inadequate dietary intake promotes the development of osteoporosis [65]. The bones of animals with Mg deficiency are delicate and easily broken, with visible trabecular microcracks, and their mechanical properties are severely degraded. Consequently, Mg deficiency has a negative effect on peri-implant cortical bone, markedly decreasing tibial cortical thickness [66]. Hypomagnesemia caused by Mg deficiency is corrected by the release of Mg from the bone surface. The newly formed hydroxyapatite crystals are larger, which contributes to bone stiffness [67]. The Framingham Heart Study [68] did not observe any association between Mg intake and BMD over the four-year period; however, higher trochanteric BMD for every 100 mg of Mg consumed by women at the beginning was reported. Orchard et al. [69] informed about decreased baseline total hip and whole body BMD in postmenopausal women with lower daily Mg intake. These authors did not notice any relationship between lower Mg intake and a higher risk of total or hip fractures.

Magnesium deficiency has an impact on bones by affecting the main regulators of Ca homeostasis―PTH and 1,25(OH)_2_ D_3_. Hypomagnesaemia prevents the release of PTH and possibly also reduces sensitivity to circulating PTH in target organs, thus producing a biochemical pattern identical with primary hypoparathyroidism [70]. Supplementation of Mg has been shown to correct the levels of PTH and 1,25(OH)_2_ D_3_ in osteoporotic postmenopausal women [71]. Moreover, the level of 1,25(OH)_2_ D_3_ in diabetic patients with low Mg levels returns to normal after Mg supplementation [72].

As recent studies show, Mg deficiency triggers an inflammatory response, which results in the activation of leukocytes and macrophages, the release of inflammatory cytokines and acute-phase proteins, and excessive free radical production [73,74]. The inflammatory response mechanisms in Mg deficiency probably include: 1) the influx of Ca into cells and the priming of phagocytic cells; 2) the opening of Ca channels and the activation of NMDA receptors; 3) the release of neurotransmitters, such as substance P; and 4) the oxidation of the membrane and activation of the nuclear factor kappa-light-chain-enhancer of activated B cells (NF-κB) [75]. Released proinflammatory cytokines play a potentially critical role in both the normal process of bone remodeling and in the pathogenesis of perimenopausal and late-life osteoporosis [76]. Interleukin (IL)-6 promotes the differentiation and activation of osteoclasts. IL-1 is another strong stimulator of bone resorption that entails the accelerated bone loss observed in idiopathic and postmenopausal osteoporosis [77]. Tumor necrosis factor-α (TNF-α) is involved in tumor-induced bone resorption and non-tumor-induced osteopenia [78]. Bone resorption and spinal bone loss in healthy pre- and postmenopausal women was positively correlated with the IL-1, IL-6, and/or TNF-α production by peripheral blood monocytes [79]. Additionally, nitric oxide (NO), the inflammatory mediator, plays a role in the pathogenesis of osteoporosis. The activation of the inducible NO synthesis (iNOS) pathway by various cytokines (e.g., IL-1 and TNF-α) inhibits osteoblast function *in vitro* and induces osteoblast apoptosis [80].

Elevated Mg levels may also have a negative effect on bone stiffness and strength [65]. Increased Mg intake in postmenopausal women was found to be accompanied by more frequent wrist fractures [81]. These findings correspond with data showing that a higher concentration of Mg may have deleterious effects on bone metabolism and parathyroid function, resulting in defective mineralization. Undoubtedly, high Mg levels in bone inhibit hydroxyapatite crystal formation by competing with Ca and binding to pyrophosphate to form an insoluble salt that is not degraded by enzymes [82]. Moreover, Mg is an antagonist of Ca. It can be assumed that high Mg levels change the Ca/Mg ratio, consequently leading to the dysregulation of cell function. Hence, an *in vitro* inhibitory effect of high Mg levels on osteoblast differentiation and mineralization has been demonstrated [64].

Magnesium plays a key role in mineral and bone homeostasis, bone cell function, as well as growth and formation of hydroxyapatite crystals. About 60% of total Mg is stored in bones, where it forms a fixed and dynamic pool [83]. This dynamic pool can be regarded as a quick exchangeable store of Mg, capable of restoring serum Mg in the case of its deficiency. This pool decreases with aging from 50% in early adolescence to 33% in adults, and then to about 10% in the elderly [83]. The highest Mg levels in bones were found in compact bone, either on the surface of hydroxyapatite or in the hydration shell around the crystal [65].

The concentration of Mg in bones is determined by factors such as age, sex, physical activity, and smoking. In the bodies of elderly people (over 60 years of age), Mg levels decrease to 60–80% of those found in children’s tissues [62]. Zaichick and Zaichick [54] reported a higher level of Mg in the bones of patients aged between 15 and 35 than between 35 and 55. Based on Table 2, it was noted that women and men have similar Mg concentrations in the bones; in some studies, women had higher Mg content in the bones than men [38,39,46,47,52]. However, it is important at what age the studied female patients were examined. During the childbearing period, pregnancy, and breastfeeding, the demand for Mg doubles. This is due to the needs of the growing fetus, the placenta, and the increase in the body weight of the pregnant woman [62]. In a study of women aged 45–55 years, a higher Mg intake was associated with higher forearm (but not femoral neck and hip) BMD. However, this relationship was not observed in women who had recently undergone menopause [24]. A cross-sectional study demonstrated a significant correlation between Mg intake and the hip BMD in 69–97-year-old men [84]. These epidemiological studies link dietary Mg intake with BMD, but indicate that there is a need for further well-designed research that would include both women and men. The study conducted in Wielkopolska voivodeship, Poland, demonstrated that the lack of regular physical activity led to a decrease in Mg levels in the femoral neck samples [85]. Smokers have lower Mg levels in the femoral head and in the spongy bone of the femur than nonsmokers [11,42,48]. Changes in Mg concentrations in the body can also be caused by alcohol and coffee abuse, diet, stress, and in the course of some diseases, including heart failure, kidney diseases, atherosclerosis, neoplastic diseases, hypertension, diabetes, and postmenopausal osteoporosis [62,86,87]. Kuo et al. [46] found an influence of seafood consumption on the Mg bone content. A positive correlation between a diet containing seafood and the level of Mg in the bone was reported. Taking into account the bone diseases, Karaaslan et al. [53] reported a higher level of the element in the femoral neck of patients with osteoarthritis than patients with fractures. Additionally, Brodziak-Dopierała et al. [36] noted that patients with femoral neck fractures had lower Mg levels than subjects with degenerative changes. This could be related to the fact that there is a strong positive correlation between Ca and Mg in the mineralized tissues [88].

The level of Mg varied depending on the type of bone and the geographical region. The highest concentration was noted in the ribs and the lowest in the tibia. The Mg concentration can be arranged: Asia> North America> Europe. The highest number of studies were conducted in Europe. We found no research in South America and Africa (Table 2).

## 4. Phosphorus (P)

Phosphorus is a macroelement involved in many biological processes. Due to its mobility, it is a key human intracellular anion, which participates in maintaining the acid–base balance in the body, creating buffer systems in blood and urine [89]. This element is part of nucleic acids and high-energy compounds, including adenosine 5'-triphosphate (ATP), adenosine 5'-diphosphate (ADP), and phosphocreatine. It is a component of phospholipids and biological membranes [90]. Phosphorus is involved in the conduction of nerve stimuli. Hydroxyapatites and phosphoproteins are bone building materials, while pyrophosphates play a regulatory role in the processes of osteogenesis and osteolysis [91].

Phosphorus is the second, next to Ca, basic component of bone tissue. In the human body, it is present in the amount of 550–770 g, almost 85% of which is stored in bones and teeth in the form of phosphoproteins and hydroxyapatite crystals [90]. Phosphorus homeostasis is regulated by three main hormones: PTH, 1,25(OH)_2_ D_3_, and FGF23 that is secreted by osteocytes [25]. The appropriate level of inorganic P is crucial for the activity of osteoblasts and osteocytes in the process of matrix mineralization [92]. The dietary Ca:P ratio is important for proper bone formation. It is currently believed that it should be 1:1 (molar ratio) / 1.3:1 (in weight units) [93]. The deficiency of P leads to mineral deposition defects, and to the presence of unmineralized osteoid, characterizing bone disorders, rickets and stunted growth in children and osteomalacia in adults. Phosphorus deficiency in the diet is very rare in humans, and most of its cases are due to defective P reabsorption in the kidneys [89]. There are many reports that high P intake negatively affects bone health in people with an extremely low dietary Ca:P ratio [94,95]. The mechanism of decreased bone formation associated with increased resorption and worse biomechanical properties due to a phosphorus-rich diet is possibly related to elevated PTH levels [25]. Nevertheless, high P consumption does not adversely affect the Ca balance in people with proper Ca and P intake [96,97,98]. *In vitro* research conducted by Kanatani et al. [99] showed that high P levels inhibited osteoclast differentiation and activity. At the same time, Fenton et al. [100] found no association between high P intake and bone demineralization. In large population studies in the US [101] and South Korea [102], no association was observed between P consumption and BMD, BMC or osteoporosis. Similarly, in perimenopausal women there was usually no between P intake from foods and BMD or BMC (however, a higher dietary Ca and the Ca:P ratio were positively associated with BMD and BMC) [103]. In contrast, in a Brazilian cohort with a low average Ca intake (~ 400 mg/d), a fracture risk increased by 9% for each 100 mg/d increase in dietary P [104]. A study of nearly 10,000 people, conducted in Korea, showed that the lower the Ca:P ratio in the diet, the greater the risk of osteoporosis (defined on the basis of BMD assessment) [105].

Animal studies show that high P levels in the diet, especially a low-calcium diet, reduces BMD due to secondary hyperparathyroidism [89]. As indicated by Draper et al. [106], parathyroidectomy prevented increased bone resorption in adult rats on a phosphorus-rich diet. The bone matrix protein, osteopontin (OPN), has recently been identified as another factor mediating the increased bone resorption in response to a diet with an excess of P content. In the study by Koyama et al. [107], four-month-old OPN-deficient and wild-type adult mice were fed a diet containing 0.5% Ca and either 0.16% or 0.6% P for four weeks. OPN deficiency prevented a decrease in BMD, changes in the pattern of trabecular bone, as well as the image and area of cortical bone, its cross-section, periosteal circumference and thickness.

Hormones are the main factors that modify the concentration of P in the blood. The absorption of P in the intestine is enhanced by PTH, calcitriol, 1,25(OH)_2_ and growth hormone. They also affect the bone turnover of P. The absorption of P is inhibited by Ca ions and by some drugs that neutralize gastric contents and bind phosphorus (e.g., aluminum hydroxide and magnesium hydroxide used to treat gastric hyperacidity).

The concentration of phosphorus varied depending on biological and environmental factors [108]. Zaichick and Zaichick [54] found statistically higher P levels in the iliac crest of women than in men. However, in most studies P concentration in the bones was at a similar level between both genders. The studies concerning the influence of age on the P content in the bones differ. Zaichick and Zaichick [54] noted higher P concentration in the iliac crest of patients aged 15–35 than 35–55, whereas Brodziak-Dopierała et al. [43] found lower P levels in the spongy femoral head bone in the 59-year-old patients than in patients aged 60–69. However, most of the studies did not reveal age-dependent differences in P levels in the bone tissue (Table 3). Based on available data, the P concentration in the bones does not differ in accordance with the place of residence. There is no difference in P levels between the residence of villages, and small or large cities [42,85,109]. It was found that alcohol consumption has no influence on the P concentration in the bones [42,109], whereas smoking habit does. Brodziak-Dopierała et al. [11], Jurkiewicz et al. [48], and Ciosek et al. [109] observed higher P concentration in the bone tissue of nonsmoking patients than smoking patients. Based on Table 3, phosphorus levels in different bones range from 9400 to 120,000 mg kg^−1^ dw with the highest level in the tibia and the lowest in the femur.

## 5. Fluorine (F)

According to some researchers, fluorine is a microelement needed for proper development. However, in the case of this element, it is important to determine its concentration in the human body, because the difference between the tolerated dose and the toxic dose is very small [110].

Fluorides (F^-^), the ionic form of fluorine, in trace amounts are essential for the proper development of the bones and the teeth [111]. They stimulate the proliferation of osteoblasts and inhibit the activity of osteoclasts, thus leading to an increase in bone mass [112]. Therefore, fluorine compounds are used in the treatment of osteoporosis. It has been observed that in the course of steroid therapy, the administration of low doses of F^-^ and 1,25(OH)_2_ D_3_ reduces the risk of vertebral fractures and protects against bone loss in these patients [113]. It has been found that F, especially in drugs with a slow release of a fluoride anion, in daily doses of 10–20 mg, may increase trabecular bone density [114].

Fluorine directly affects the bones by two main mechanisms. In mineralized tissues, F is incorporated into apatite crystals in the process of ion exchange, which leads to the formation of fluorapatite [111]. Such conversion results in changes in crystallinity and a reduction in mechanical properties [115]. Fluorapatite is more stable and less soluble in acids, which may result in a higher resistance to bone resorption by osteoclasts. It turns out, however, that its accretion is perpendicular to the collagen fibers, unlike hydroxyapatite. It is also less conducive to binding with proteins. Fluorine also stimulates bone formation by inducing proliferation and activity of osteoblasts. No direct effect of F on osteoclasts has been reported [116]. The ionic form of F stimulates the proliferation of bone cells by directly inhibiting the phosphotyrosyl protein phosphatase activity. This increases total cellular tyrosyl phosphorylation, thus leading to the stimulation of bone cell proliferation [117]. Fluoride also increases the level of bone cell growth factor, such as insulin-like growth factor–1 (IGF-1) and osteoblastic transforming growth factor-β1 (TGF-β1) [118]. The osteogenic action of F^-^ has been suggested to involve the activation of the mitogen-activated protein kinase (MAPK) [111,119]. The administration of F^-^ in rats increased expression of mRNA of collagen type I alpha 1 chain (COL1A1), alkaline phosphatase (ALP) and runt-related transcription factor 2 (Runx2), which could be blocked by Dickkopf-related protein 2 (DKK-2), an inhibitor of the Wnt/β-catenin receptor. Thus, fluoride stimulates osteoblastogenesis by the canonical Wnt pathway [120,121].

Excessive fluoride intake causes skeletal fluorosis―a condition characterized by radiographic bone changes ranging from osteoporosis to osteosclerosis [122]. Histomorphometric examinations in fluorosis revealed an increase in the number of osteoblasts, in the density of the spongy bone, in the thickness of the trabeculae, and in the volume of the osteoid. The typical features of bone under the influence of fluoride are macular osteocytic sinuses, resulting from a delay in mineralization. There is also a widening of and an increase in the porosity of the cortical layer, and increased periosteal activity [123]. Excessive fluoride intake may disturb bone turnover, thus affecting the differentiation of osteoblasts and osteoclasts, and induce the development of bone changes [124,125]. This ultimately results in an imbalance between bone formation and bone resorption [111,126,127]. It is suggested that the detrimental effect of fluoride on the skeleton is caused by an overproduction of PTH and activated bone resorption [128,129]. Exposure to F may also modulate PTH secretion, and thereby changes in Ca levels. Zierold and Chauviere [130] documented decreased serum Ca levels after acute hydrofluoric acid inhalation.

In the human body, 93-97% of fluorine is stored in hard tissues, and the rest is accumulated in the organs, including the liver and the kidneys [131]. The highest F levels are observed on the bone surface. The concentration of F in trabecular bone is approximately two to three times higher than in cortical bone. As for cortical bone, the highest F levels are usually found on its surface [127]. The spongy bone was found to be more resistant to the effect of F than the compact bone. In children, F retention in bones is greater than in adults [132]. It has been found that children and adults exposed to low doses of F compounds accumulate approximately 50% and 10% of the F taken, respectively, in the bone tissue [133]. The half-life of F in hard tissues ranges from several to even 20 years [134]. There are a few studies concerning F concentration in the bones. The accumulation of F in bones is determined, among others, by the duration of exposure, age, sex, as well as bone diseases [135,136]. Higher F is accumulated in female than in male bones [137]. Fluoride gradually accumulates in the bone through life. Higher levels of F were found in patients over 60 years old than below 60 years of age [137,138,139,140]. Based on the available data presented in Table 4, the F concentration in the bones is related to contact with fluoridated water supply or preparations containing F.

Based on the analysis of the concentration of F^-^ in different types on bones, it was noticed that F- levels were highest in the vertebrae (>500 mg kg^−1^ dw), ribs (100–500 mg kg^−1^ dw) and the femur (100–450 mg kg^−1^ dw), and lower in the long bones (100–300 mg kg^−1^ dw) and the other bones (>120 mg kg^−1^ dw) [136], (Table 4). Physiological F levels in bones were found to be <550 mg kg^−1^ dw. The reference value of F accumulated in bones should be kept below 0.4–0.5% bone ash [141]. Table 4 presents the literature data concerning F levels.

## 6. Lead (Pb)

Lead is one of the elements that do not play any physiological role in the human body. Lead has an impact on the activity of many enzymes and the function of structural proteins [146]. This element has been found to influence the activity of the enzymes involved in oxidative phosphorylation, glycolysis, and heme synthesis, disturbing cellular metabolic changes, including the regulation of energy processes, protein and nucleic acid synthesis [146,147]. Lead can combine with the proteins responsible for the neutralization of free radicals, thus leading to oxidative stress [148,149].

Lead is highly cytotoxic. It affects osteoclasts and osteoblasts. Additionally, it can influence the metabolism of 1,25(OH)_2_ D_3_, and therefore it has been recognized as a factor contributing to osteoporosis, especially in perimenopausal women [150,151]. The effect of Pb on bone cells can be both indirect and direct. Lead indirectly impairs the hydroxylation of 1,25(OH)_2_ D_3_ in the kidneys. It shows a high affinity for thiol groups in the active sites of some enzymes. Poisoning with Pb influences the mineral metabolism of Ca and P by inhibiting the renal 1-α-hydroxylase enzyme, which is necessary for the synthesis of 1,25(OH)_2_ D_3_. This results in low levels of Ca and P due to the decreased intestinal absorption of these minerals [152].

Animal studies show that Pb directly reduces BMD and cortical bone width, increases susceptibility to fractures, and impairs the union of fractured bone [153]. Additionally, Pb causes an imbalance in bone remodeling, enhancing both bone formation (increase in the amino-terminal propeptide (PINP) of type I collagen) and resorption (increase in carboxy-terminal telopeptide of type I collagen (CTX)). The processes of bone formation and mineralization are accelerated, which results in the formation of poor quality bones [153].

*In vitro* studies have shown that Pb restrains osteoblast activity by inhibiting Wnt/β-catenin signaling, a critical anabolic pathway for osteoblastic bone formation [154], and that it can induce apoptosis in osteoblasts [155]. As suggested by Ma et al. [155], one of the mechanisms of lead-induced apoptosis may be the activation of intracellular Ca stores by reducing the protein level and the enzymatic activity of phosphatidylcholine-specific phospholipase C (PC-PLC), which may increase Ca^2+^ levels, and thus the apoptotic signal pathway can be induced. Al-Ghafari et al. [156] reported the Pb level threshold of 0.1 μM for osteoblast death *in vitro* after exposure of 48 h. At the level of 1 μM, cytotoxic effects were apparent already after six hours. Additionally, disturbed osteoblast bioenergetics were observed. The potential of the mitochondrial membrane was reduced, and the activity of the mitochondrial proteins was disrupted and inhibited. The authors also observed elevated levels of reactive oxygen species (ROS) within osteoblasts in response to 24-h exposure to Pb (55 μM), cellular redox stress aggravated by a reduced activity of superoxide dismutase (SOD) and catalase (CAT), as well as decreased levels of cellular glutathione [156]. Additionally, the activity of nuclear factor erythroid 2-related factor 2 (NRF2) significantly increased as a response to toxic lead levels. The authors assumed that the increased NRF2 activity was to alleviate redox stress [156].

Lead inhibits the synthesis of osteocalcin, which promotes bone mineralization and density, and plays a role as a bone-derived hormone that influences energy metabolism. Moreover, it has been found to influence the level of alkaline phosphatase (ALP), which is one of the components of the organic bone matrix [157]. Based on their *in vitro* analysis, Al-Ghafari et al. [156] observed that the production of osteocalcin and ALP reduced after treating osteoblasts with Pb.

The principal target for Pb is the bone matrix because of its ability to replace other divalent cations in the body. Dowd et al. [158] proposed a pathway for the molecular mechanism of Pb. Lead shows a greater affinity for osteocalcin than calcium. Conformational changes induced by Ca in the protein improve its binding propensity to hydroxyapatite, and so Pb can replace Ca in the hydroxyapatite crystal. According to Dowd et al. [158], replacing Ca with Pb, either in the complex with osteocalcin or in hydroxyapatite, would improve the binding rate of the protein to the crystal. Even though the role of osteocalcin is not clear, the result may be changes in bone remodeling.

Lead is accumulated in the liver, lungs, heart and kidneys as a quick exchange pool, in the skin and muscles as an intermediate exchange pool, and in bone tissue as a slow exchange pool [159]. The main reservoir of this element in the body is bone tissue, where it occurs in the form of colloidal and crystalline compounds, mainly in the areas of intense bone mineralization. The concentration of Pb varies depending on the type of bones. Based on available data, the highest level of Pb is found in the femoral head and the lowest in the femoral neck, which can be explained by the fact that Pb is transported via blood and the femoral head is more vascularized than the femoral neck [160]. As documented in the literature, the lead levels in various bone elements range from 0.85 to 15.11 mg kg^−1^ dw (Table 5). The concentration of Pb in the tissues changes during life. Bone Pb levels in adults increase with age by up to ten times, especially in the tibias [161]. In adults 90–95% of Pb is stored in the spongy and cortical bones [162], with a higher level in the cortical bone (Table 5), while in children it is suggested that most Pb (70–95%) accumulates in the trabecular bone due to its high turnover rate [162]. The biological half-life of Pb in the trabecular bone is estimated to be one year, and in the cortical bone from 10–20 years [152]. Based on the data presented in Table 5, the Pb concentration is higher in males than females. It is suggested that in females during or after the menopause, a rise of Pb in the blood stimulates its accumulation in other organs [152,163], and most studies were conducted on patients above 50 years of age. The accumulation of Pb in the bones is also determined by the duration of exposure, smoking, alcohol abuse, diet, and some diseases. Higher levels of Pb in the bones are found in smoking patients than in nonsmokers [35,48,164,165]. Taking into account the influence of diet on Pb cumulation in the human body, Cirillo et al. [166] and Pastorelli et al. [167] found that seafood represents a non-negligible contribution to Pb intake, but the exposure to Pb with seafood does not exceed the standard tolerable weekly intake. However, Kuo et al. [46] found a positive correlation between the frequency of seafood consumption and Pb concentration in the bones. Based on Table 5, it is noted that Pb level in patients from places where consumption of seafood is more frequent, is ~5 times higher than in patients from places where consumption of seafood is occasional. In comparison, we included China, Taiwan, Spain, and Japan as places where consumption of seafood is more frequent and northwestern part of Poland where consumption of seafood is occasional. We did not take into account the upper Silesia part of Poland due to the fact that this region is the most industrialized area in Poland, and other factors influence the Pb concentration. Łanocha et al. [168] analyzed the influence of fish consumption on the Pb levels in the spongy bone of the femoral head of patients from the northwestern part of Poland. The authors found higher Pb content in patients consuming fish and seafood several times a month than in patients who consume fish and seafood once a month. At the same time, Łanocha et al. [168] found much higher Pb level in the bones of patients who did not consume fish or seafood, but there were only 4 subjects in the group and other factors might influence the results of this group. More details are presented in Table 5.

## 7. Conclusions

While calcium, magnesium, and phosphorus are crucial for bone tissue homeostasis, fluorine should be approached with caution. In low concentrations, fluorine has an important impact on bone tissues, but its excess in the body can damage bone integrity. Lead is toxic to bone cells even at low concentrations. Our review shows the effect of these elements on bones. Researchers inform about important relationships between calcium, magnesium, phosphorus, and lead. An increase or decrease in the level of one element may have an effect on the action of the other one. Despite numerous advances in the knowledge, much remains to be elucidated about the impact of heavy metals (including lead) on bone tissue. The understanding of their mechanism of action will allow finding treatments to counteract their negative effect on bone.

## Figures and Tables

**Table 1 biomolecules-11-00506-t001:** Calcium levels in human bones in mg kg^−1^ dw. (F, female; M, male; N, number of participants; NS, nonsmoking; S, smoking).

The Studied Area	Age	Sex	N	Ca Level	Additional Information	Reference
**Vertebra**						
USA, Philadelphia	48	M	1	290,100.00	Chronic F poisoning	[37]
**Ribs**						
Japan, Tokyo	61–96	F + M	45	246,000.00		[38]
		F	28	245,000.00		
		M	17	248,000.00		
Russia, Obninsk	15–55	F + M	80	171,395.00		[39]
		F	38	182,183.00		
		M	42	161,397.00		
USA, Philadelphia	48	M	1	292,400.00	Chronic F poisoning	[37]
**Ribs (Spongy Bone)**						
USA, Kentucky	69 ± 6.3	F + M	12	190,000.00		[40]
		F	4	-		
		M	8	-		
**Ribs (Cortical Bone)**						
USA, Kentucky	60–82	F + M	12	210,000.00		[40]
**Femur**						
Poland, Upper Silesia	67.5	F + M	50	107,000.00		[41]
	67.2	F	36	104,200.00		
	68.1	M	14	122,500.00		
USA, Philadelphia	48	M	1	293,700.00	Chronic F poisoning	[37]
**Femoral Head**						
Poland, Greater Poland Voivodeship	63.8	F + M	96	136,705.60		[42]
	64.5 ± 14.2	F	57	132,738.00		
	63.2 ± 10.2	M	39	142,503.00		
Poland, Upper Silesia	71 ± 6	F	64	28,512.42		[43]
		M	39	29,099.37		
Poland, Upper Silesia	65.6	F + M	53	49,485.45		[44]
Poland, Silesia, Łódź, Cracow	65.8 ± 12.5	F + M	197	170,100.00		[45]
Taiwan	-	F + M	70	82,007.90		[46]
	41–60	F	17	77,115.00		
	61–80	M	53	84,649.00		
**Head of the Femur (Spongy Bone)**						
Poland, Upper Silesia	71 ± 6	F + M	103	25,244.65		[43]
Poland, Upper Silesia	68 ± 9.9	F + M	13	174,400.00	None of the patients had ever been occupationally exposed to heavy metals	[47]
		F	9	177,200.00		
		M	4	168,900.00		
Poland, Łódź	68.3 ± 7.3	F + M	12	169,500.00	None of the patients had ever been occupationally exposed to heavy metals	[47]
		F	10	170,600.00		
		M	2	164,100.00		
Poland, Cracow	69.2 ± 9.6	F + M	13	207,000.00	None of the patients had ever been occupationally exposed to heavy metals	[47]
		F	10	198,600.00		
		M	3	234,700.00		
Poland, Lower Silesian Voivodeship	65.9 ± 10.8	F + M	21	159,900.00	NS	[48]
	62.8 ± 17.2		22	156,000.00	S	
Poland, Upper Silesia	77 ± 5	F + M	110	27,929.17		[49]
Poland, Upper Silesia	65.6	F + M	53	-		[44]
	67.2	F	43	160,730.00		
	58.4	M	10	160,120.00		
Poland, Upper Silesia	65.7	F + M	91	127,480.00		[36]
China, Shanghai	62	F	1	97,700.00		[50]
Poland, West Pomeranian Voivodeship	52–84	F + M	37	243,700.00		[35]
	69.90 ± 77	F	24	242,040.00		
	62.6 ± 15.4	M	13	249,460.00		
Poland, West Pomeranian Voivodeship		F + M	29	263,990.00	A diet exclusive of game meat	[35]
			8	228,440.00	A diet inclusive of game meat	
Poland, Upper Silesia	71.6	F + M	103	30,216.83	Patients living in the industrial area	[51]
		F	69	28,512.42		
		M	39	29,099.37		
Poland, Upper Silesia	65.7	F + M	91	43,520.00		[40]
Poland, Upper Silesia	65.6	F + M	53	39,460.00		[44]
	67.2	F	43	48,970.00		
	58.4	M	10	49,610.00		
Poland, Upper Silesia	77.0 ± 5.0	F + M	110	25,212.61		[49]
China, Shanghai	62	F	1	216,500.00		[50]
Poland, West Pomeranian Voivodeship	52–84	F + M	37	225,520.00	People with osteoarthritis	[35]
	69.90 ± 77	F	24	225,520.00		
	62.6 ± 15.4	M	13	223,830.00		
Poland, West Pomeranian Voivodeship		F + M	29	212,000.00	A diet exclusive of game meat	[35]
			8	267,440.00	A diet inclusive of game meat	
**Head of the Femur (Cortical Bone)**						
Poland, Upper Silesia	71 ± 6	F + M	103	30,216.83		[43]
**Femoral Neck**						
Poland, Greater Poland Voivodeship	63.8	F + M	96	157,212.30		[42]
	64.5 ± 14.2	F	57	158,841.00		
	63.2 ± 10.2	M	39	154,831.00		
Russia, Obninsk	15-55	F	38	158,000.00		[52]
		M	47	149,500.00		
Turkey, Erciyes	73.9 ± 9.7	F + M	30	9.7	Fracture group Osteoarthritis	[53]
	72.8 ± 6.0					
Turkey, Erciyes	60.5 ± 5.1	F + M	30	9.7	Fracture group	[53]
	63.2 ± 6.6					
**Tibia**						
Poland, Upper Silesia	67.5	F + M	50	124,500.00		[41]
	67.2	F	36	121,900.00		
	68.1	M	14	131,700.00		
Russia, Obninsk	15-55	F	38	176,000.00	Within 24 h of death, healthy humans	[54]
	15-55	M	46	164,000.00		
	15-35	F	-	182,000.00		
	15-35	M	-	173,000.00		
	15-35	F + M	-	177,000.00		
	36-55	F	-	168,000.00		
	36-55	M	-	157,000.00		
	36-55	F + M	-	162,000.00		
USA, Philadelphia	48	M	1	293,200.00	Chronic F poisoning	[37]

**Table 2 biomolecules-11-00506-t002:** Magnesium levels in human bones in mg kg^−1^ dw. (F, female; M, male; N, number of participants; NS, nonsmoking; S, smoking).

The Studied Area	Age	Sex	N	Mg Level	Additional Information	Reference
**Ribs**						
Japan, Tokyo	61–96	F + M	45	2,850.00		[38]
		F	28	2,920.00		
		M	17	2,730.00		
Russia, Obninsk	15–55	F + M	80	2,139.00		[39]
		F	38	2,218.00		
		M	42	2,067.00		
**Ribs (Spongy Bone)**						
USA, Kentucky	69 ± 6.3	F + M	12	2,700.00		[40]
**Ribs (Cortical Bone)**						
USA, Kentucky	60–82	F + M	12	2,500.00		[40]
**Femur**						
Poland, Upper Silesia	67.5	F + M	50	1,443.62		[41]
	67.2	F	36	1,437.11		
	68.1	M	14	1,471.24		
**Femoral Head**						
Poland, Greater Poland Voivodeship	63.8	F + M	96	1,446.76		[42]
	64.5 ± 14.2	F	57	1,415.20		
	63.2 ± 10.2	M	39	1,492.90		
Poland, Upper Silesia	71 ± 6	F	64	1,352.49		[43]
		M	39	1,818.91		
Poland, Upper Silesia	65.6	F + M	53	1,306.20		[44]
Poland, Silesia, Łódź, Cracow	65.8 ± 12.5	F + M	197	1,765.00		[45]
Taiwan	-	F + M	70	3,005.20		[46]
	41–60	F	17	3,304.00		
	61–80	M	53	2,843.00		
**Femoral Head (Spongy Bone)**						
Poland, Upper Silesia	68 ± 9.9	F + M	13	1,813.30	None of the patients had ever been occupationally exposed to heavy metals	[47]
		F	9	1,874.90		
		M	4	1,689.90		
Poland, Łódź	68.3 ± 7.3	F + M	12	1,792.90	None of the patients had ever been occupationally exposed to heavy metals	[47]
		F	10	1,824.00		
		M	2	1,637.40		
Poland, Cracow	69.2 ± 9.6	F + M	13	2,032.00	None of the patients had ever been occupationally exposed to heavy metals	[47]
		F	10	1,996.70		
		M	3	2,149.50		
Poland, Silesian Voivodeship	65.9 ± 10.8	F + M	21	1,798.47	NS	[48]
	62.8 ± 17.2	F + M	22	1,614.61	S	
Poland, Upper Silesia	77 ± 5	F + M	110	873.22		[49]
Poland, Upper Silesia	71 ± 6	F + M	103	1,650.85		[43]
Poland, Upper Silesia	65.7	F + M	91	3,040.00		[36]
**Femoral Head (Cortical Bone)**						
Poland, Upper Silesia	71.6	F + M	103	1,376.14		[51]
		F	69	1,352.49		
		M	39	1,818.91		
Poland, Upper Silesia	65.7	F + M	91	910.00		[36]
Poland, Upper Silesia	77 ± 5	F + M	110	1,028.59		[49]
Poland, Upper Silesia	71 ± 6	F + M	103	1,376.14		[43]
**Femoral Neck**						
Poland, Greater Poland Voivodeship	63.8	F + M	96	1,585.80		[42]
	64.5 ± 14.2	F	57	1,599.30		
	63.2 ± 10.2	M	39	1,566.10		
Russia, Obninsk	15–55	F	38	2,058.50		[52]
		M	47	1,850.50		
Turkey, Erciyes	73.9 ± 9.7	F	30	2,183.30	Patients with fractures	[53]
	72.8 ± 6.0	M				
Turkey, Erciyes	60.5 ± 5.1	F	30	2,566.70	Patients with osteoarthritis	
	63.2 ± 6.6	M				
**Tibia**						
Poland, Upper Silesia	67.5	F + M	50	1,572.40		[41]
	67.2	F	36	1,562.44		
	68.1	M	14	1,591.00		
**Iliac Crest**						
Russia, Obninsk	15–55	F	38	1,995.00	Within 24h of death, healthy humans	[54]
	15–55	M	46	1,710.00		
	15–35	F	-	2,138.00		
	15–35	M	-	1,904.00		
	15–35	F + M	-	2,009.00		
	36–55	F	-	1,867.00		
	36–55	M	-	1,536.00		
	36–55	F + M	-	1,687.00		

**Table 3 biomolecules-11-00506-t003:** Phosphorus levels in human bones in mg kg^−1^ dw. (F, female; M, male; N, number of participants; NS, nonsmoking; S, smoking).

The Studied Area	Age	Sex	N	P Level	Additional Information	Reference
**Vertebra**						
USA, Philadelphia	48	M	1	113,700.00	Chronic F poisoning	[37]
**Ribs**						
Japan, Tokyo	61–96	F + M	45	119,000.00		[38]
		F	28	120,000.00		
		M	17	119,000.00		
Russia, Obninsk	15–55	F + M	80	75,137.00		[39]
		F	38	78,481.00		
		M	42	72,037.00		
USA, Philadelphia	48	M	1	115,100.00	Chronic F poisoning	[37]
**Ribs (Spongy Bone)**						
USA, Kentucky	69 ± 6.3	F + M	12	96,000.00		[40]
		F	4			
		M	8			
**Ribs (Cortical Bone)**						
USA, Kentucky	60–82	F + M	12	95,000.00		[40]
**Femur**						
Poland, Upper Silesia	67.5	F + M	50	48,400.00		[41]
	67.2	F	36	47,000.00		
	68.1	M	14	55,100.00		
USA, Philadelphia	48	M	1	116,000.00	Chronic F poisoning	[37]
**Femoral Head**						
Poland, Greater Poland Voivodeship	63.8	F + M	96	62,723.00		[42]
	64.5 ± 14.2	F	57	60,738.00		
	63.2 ± 10.2	M	39	65,624.00		
**Femoral Head (Spongy Bone)**						
Poland, Upper Silesia	68 ± 9.9	F + M	13	61,600.00	None of the patients had ever been occupationally exposed to heavy metals	[47]
		F	9	65,900.00		
		M	4	52,800.00		
Poland, Łódź	68.3 ± 7.3	F + M	12	58,700.00	None of the patients had ever been occupationally exposed to heavy metals	[47]
		F	10	59,100.00		
		M	2	56,400.00		
Poland, Cracow	69.2 ± 9.6	F + M	13	73,800.00	None of the patients had ever been occupationally exposed to heavy metals	[47]
		F	10	65,500.00		
		M	3	101,400.00		
China, Shanghai	62	F	1	49,900.00		[50]
**Femoral Head (Cortical Bone)**						
China, Shanghai	62	F	1	110,000.00		[50]
**Femoral Neck**						
Poland, Greater Poland Voivodeship	63.8	F + M	96	70,652.30		[42]
	64.5 ± 14.2	F	57	71,697.00		
	63.2 ± 10.2	M	39	69,124.00		
Russia, Obninsk	15–55	F	38	75,850.00		[52]
		M	47	70,850.00		
**Tibia**						
Poland, Upper Silesia	67.5	F + M	50	55,800.00		[41]
	67.2	F	36	55,300.00		
	68.1	M	14	59,600.00		
USA, Philadelphia	48	M	1	114,400.00	Chronic F poisoning	[37]
**Tibia (Spongy Bone)**						
Poland, West Pomeranian Voivodeship	-	F + M	46	72,471.09		[109]
	73.1 ± 8.4	F	34	72,845.40		
	73.6 ± 8.7	M	12	71,663.26		
	55–74	F + M	26	73,880.72		
	75–89	F + M	20	73,171.86		
		F + M	2	75,990.78	Normal weight	
		F + M	19	73,917.05	Overweight	
		F + M	17	70,315.04	Class I obesity	
		F + M	6	72,220.95	Class II obesity	
		F + M	2	78,862.89	Class III obesity	
		F + M	5	65,086.30	Rural areas	
		F + M	6	53,621.20	Up to 100,000 inhabitants	
		F + M	35	65,490.03	Over 100,000 inhabitants	
		F + M	6	70,315.04	S	
		F + M	40	72,572.29	NS	
		F + M	6	71,365.24	Regular alcohol	
		F + M	40	72,543.91	Consumption	
**Iliac Crest**						
Russia, Obninsk	15–55	F	38	84,500.00	Within 24h of death, healthy humans	[54]
	15–55	M	46	75,600.00		
	15–35	F	-	88,300.00		
	15–35	M	-	79,800.00		
	15–35	F + M	-	83,600.00		
	36–55	F	-	81,200.00		
	36–55	M	-	71,700.00		
	36–55	F + M	-	76,000.00		

**Table 4 biomolecules-11-00506-t004:** The ionic form of fluorine levels in human bones in mg kg^−1^ dw. (F, female; M, male; N, number of participants; NS, nonsmoking; S, smoking).

The Studied Area	Age	Sex	N	F^-^ Level	Additional Information	Reference
**Vertebra**						
USA, Philadelphia	48	M	1	7,000.00	Chronic F poisoning	[37]
Japan, Yahaba	39–79	F + M	16	100.80	Cervical vertebra	[138]
Japan, Yahaba	39–79	F + M	16	100.30	Thoracic vertebra	[138]
Japan, Yahaba	39–79	F + M	16	110.50	Lumbar vertebra	[138]
Denmark, Aarhus	21–91	F	36	1337.70		[140]
	20–82	M	37	1181.10		
**Ribs**						
USA, Philadelphia	48	M	1	5,600.00	Chronic F poisoning	[37]
Japan, Yahaba	39–79	F + M	16	99.90		[138]
Poland, Gdańsk	17–87	F + M	59	520.00		[139]
**Femur**						
USA, Philadelphia	48	M	1	2,900.00	Chronic F poisoning	[37]
Japan, Yahaba	39–79	F + M	16	140.90		[138]
**Femoral Head**						
Canada, Toronto	66 ± 11	F + M	53	1030.00	Fluoridated water supplies	[142]
Canada, Montreal	70 ± 13	F + M	39	643.00	Nonfluoridated water supplies	
**Femoral Head (Spongy Bone)**						
Poland, West Pomeranian Voivodeship	62.75	F + M	49	436.82	Patients who had been treated with preparations containing F	[143]
	66.9 ± 12.6	F	35	435.45		
	58.6 ± 11.3	M	14	501.04		
Poland, West Pomeranian Voivodeship		F + M	29	542.13	Patients who had been treated with preparations containing F	[143]
		F	21	464.42		
		M	8	765.36		
Poland, Lubuskie Voivodeship		F + M	20	387.16	Patients who had been treated with preparations containing F	[143]
		F	14	370.62		
		M	6	398.94		
**Femoral Head (Cortical Bone)**						
Poland, West Pomeranian Voivodeship	62.75	F + M	49	428.26	Patients who had been treated with preparations containing F	[143]
	66.9 ± 12.6	F	35	447.01		
	58.6 ± 11.3	M	14	393.99		
Poland, West Pomeranian Voivodeship		F + M	29	456.91	Patients who had been treated with preparations containing F	[143]
		F	21	456.91		
		M	8	456.56		
Poland, Lubuskie Voivodeship		F + M	20	359.23	Patients who had been treated with preparations containing F	[143]
		F	14	401.42		
		M	6	321.67		
**Tibia**						
North-western Poland	65.75	F + M	33	511.46		[144]
	67	F	22	513.16		
	64.5	M	11	449.31		
USA, Philadelphia	48	M	1	1,800.00	Chronic F poisoning	[37]
**Tibia (Cortical Bone)**						
Poland, north-western part	62.75	F + M	49	497.44	Patients who had been treated with preparations containing F	[143]
	66.9 ± 12.6	F	35	508.15		
	58.6 ± 11.3	M	14	449.31		
Poland, West Pomeranian Voivodeship		F + M	29	897.83	Patients who had been treated with preparations containing F	[143]
		F	21	893.84		
		M	8	1,099.95		
Poland, Lubuskie Voivodeship		F + M	20	438.34	Patients who had been treated with preparations containing F	[143]
		F	14	401.30		
		M	6	438.34		
**Tibia (Spongy Bone)**						
North-western Poland	-	F + M	20	421.36		[145]
	70 ± 10.3	F	15	458.99		
	66.3 ± 11.6	M	5	198.10		

**Table 5 biomolecules-11-00506-t005:** Lead levels in human bones in mg kg^−1^ dw. (F, female; H, hazardous waste incinerator; M, male; N, number of participants; NS, nonsmoking; S, smoking).

The Studied Area	Age	Sex	N	Pb Level	Additional Information	Reference
**Ribs**						
Spain, Tarragona	56 ± 20	F + M	78	1.79		[169]
Spain, Tarragona	<35 F >65 M	F + M	22	2.11	People who had lived for ten years near HWI	[170]
Spain	~51	F + M	20	3.99 (in 1998)	People who had lived for ten years near HWI	[171]
				2.11 (in 2003)		
				2.66 (in 2007)		
				1.39 (in 2013)		
Japan, Tokyo	61–96	F + M	45	6.85		[38]
		F	28	5.34		
		M	17	7.57		
Russia, Obninsk	15–55	F + M	80	2.24		[172]
		F	38	2.10		
		M	42	2.36		
Poland, Gdańsk	17–87	F + M	59	2.70		[139]
**Ribs (Spongy Bone)**						
USA, Kentucky	69 ± 6.3	F + M	12	5.00		[40]
**Ribs (Cortical Bone)**						
USA, Kentucky	60–82	F + M	12	13.40		[40]
**Femur**						
Poland, Upper Silesia	67.5	F + M	50	2.05		[41]
	67.2	F	36	1.74		
	68.1	M	14	2.90		
South Africa, Pretoria area	20–29	F + M	9	2.22	Black individuals who died between 1943–2012	[173]
	30–39		12	4.14		
	40–39		18	3.30		
	50–59		12	3.67		
	60–69		11	4.53		
	70–79		10	7.20		
	80–89		1	12.95		
	40–49	F + M	5	10.04	White individuals who died between 1943–2012	
	50–59		7	10.85		
	60–69		9	12.70		
	70–79		3	3.41		
	80–89		3	7.59		
	90–99		2	49.07		
**Femoral Head**						
Poland, Greater Poland Voivodeship	63.8	F + M	96	1.15		[42]
	64.5 ± 14.2	F	57	0.86		
	63.2 ± 10.2	M	39	1.57		
Poland, Upper Silesia	63.9	F + M	26	10.84	NS	[164]
				15.11	S	
Poland, Upper Silesia	71 ± 6	F	64	11.46		[43]
		M	39	10.99		
Poland, Upper Silesia	65.6	F + M	53	3.75		[44]
Poland, Silesia, Łódź, Cracow	65.8 ± 12.5	F + M	197	2.76		[45]
Taiwan	-	F + M	70	7.10		[46]
	41–60	F	17	6.76		
	61–80	M	53	6.77		
Femoral Neck						
Poland, Greater Poland Voivodeship	63.8	F + M	96	1.08		[42]
	64.5 ± 14.2	F	57	0.85		
	63.2 ± 10.2	M	39	1.41		
**Femoral Head (Spongy Bone)**						
Poland, Upper Silesia	68 ± 9.9	F + M	13	1.90	None of the patients had ever been occupationally exposed to heavy metals	[47]
		F	9	1.62		
		M	4	2.48		
Poland, Łódź	68.3 ± 7.3	F + M	12	1.35	None of the patients had ever been occupationally exposed to heavy metals	
		F	10	1.19		
		M	2	2.15		
Poland, Cracow	69.2 ± 9.6	F + M	13	2.98	None of the patients had ever been occupationally exposed to heavy metals	
		F	10	2.43		
		M	3	4.80		
Poland, Silesian Voivodeship	65.9 ± 10.8	F + M	21	2.00	NS	[48]
	62.8 ± 17.2	F + M	22	3.09	S	
Poland, Upper Silesia	63.9 ± 14.4	F + M	43	2.56		[109]
	64.87 ± 13.4	F	32	2.02		
	61.07 ± 17.5	M	11	4.12		
Poland, Upper Silesia	77 ± 5	F + M	110	2.23		[49]
Poland, Upper Silesia	71 ± 6	F + M	103	6.22		[43]
Poland, Upper Silesia	65.6	F + M	53	-		[44]
	67.2	F	43	1.75		
	58.4	M	10	1.97		
Poland, Upper Silesia	71.6	F + M	103	6.22		[51]
Poland, Upper Silesia	65.5	F + M	19	1.53	NS	[165]
			34	2.97	S	
Poland, northwestern part	66.25	F + M	37	0.50	Patients treated with arthroplasty	[174]
	69.9 ± 10.76	F	24	0.51		
	62.6 ± 15.5	M	13	0.49		
	<60	M	13	0.48		
	>60	M	13	0.47		
Poland, Szczecin	65.25	F + M	30	0.49	Patients treated with arthroplasty	[174]
	70.1 ± 10.56	F	20	0.47		
	60.41 ± 9.51	M	10	0.49		
Poland, Szczecin	64.25 ± 11.93	F + M	15	0.48	NS	[168]
			15	0.47	S fewer than 20 cigarettes per day	
			7	0.48	S more than 20 cigarettes per day	
			32	0.50	Patients without osteoporosis	
			5	0.51	Patients with Documented osteoporosis	
			4	0.53	Bones from people not consuming fish or seafood	
			15	0.43	Patients consuming fish and seafood once a month	
			18	0.49	Patients consuming fish and seafood several times a month	
China, Shanghai	62	F	1	2.41		[50]
**Femoral Head (Cortical Bone)**						
Poland, Upper Silesia	63.9 ± 14.4	F + M	36	3.05		[175]
	64.87 ± 13.4	F	26	2.55		
	61.07 ± 17.5	M	10	4.35		
Poland, Upper Silesia	65.5	F + M	34	2.13	S	[165]
			19	1.57	NS	
Poland, Upper Silesia	71.6	F + M	103	12.27	People living in the industrial area	[51]
		F	69	11.46		
		M	39	10.99		
Poland, Upper Silesia	65.6	F + M	53	1.05		[44]
	67.2	F	43	1.62		
	58.4	M	10	1.93		
Poland, Upper Silesia	77.0 ± 5.0	F+M	110	6.77		[49]
Poland, Upper Silesia	71.0 ± 6.0	F + M	103	12.27		[43]
China, Shanghai	62	F	1	4.06		[50]
Poland, Szczecin	65.25	F + M	30	0.60	Patients treated with arthroplasty	[174]
	70.1 ± 10.56	F	20	0.49		
	60.41 ± 9.51	M	10	0.83		
**Tibia**						
Poland, Upper Silesia	67.5	F + M	50	2.1		[41]
	67.2	F	36	1.73		
	68.1	M	14	3.29		
Poland, West Pomeranian Voivodeship	65.75	F + M	33	1.72		[35]
	67	F	22	1.53		
	64.5	M	11	2.06		
	-	F + M	18	2.09	S	
	-	F + M	15	1.45	NS	

## Data Availability

No new data were created or analyzed in this study. Data sharing is not applicable to this article

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
