# Peer review of "The Effects of Calcium, Magnesium, Phosphorus, Fluoride, and Lead on Bone Tissue"

_biomolecules, 2021, doi:10.3390/biom11040506_

Round 1
Reviewer 1 Report
The authors of the review described the effect of calcium, magnesium, phosphorus, fluorine and lead elements on bones. The relationships between calcium, magnesium, phosphorus, and lead are also detailed. In my opinion, it is a very interesting and useful manuscript. One question I would have:
Table 1-4 contains a lot of data from Poland, but few from other countries. Not enough data? It would be interesting to enter data from Africa, South America.
Author Response
Dear Editor,
Subject:
Thank you for your email dated 14 March 2021 enclosing the reviewers’ comments.
We have carefully reviewed the comments and have revised the manuscript accordingly. Our responses are given in a point-by-point manner below. Changes to the manuscript are shown in red.
We hope the revised version is now suitable for publication and look forward to hearing from you in due course.
Sincerely,
Dr Karolina Kot
Response to Reviewer 1:
Thank you for your review of our paper. We have answered each of your point below.
The authors of the review described the effect of calcium, magnesium, phosphorus, fluorine and lead elements on bones. The relationships between calcium, magnesium, phosphorus, and lead are also detailed. In my opinion, it is a very interesting and useful manuscript. Thank you
One question I would have: Table 1-4 contains a lot of data from Poland, but few from other countries. Not enough data? It would be interesting to enter data from Africa, South America. The studies concerning metal concentration in the bones are mostly conducted in the Europe. We have found a few new data from Europe and Asia, which we added in the tables. We have also found one article about lead concentration in the bones of patients from Africa, and we added information to the Table.

Reviewer 2 Report
The topic of this review is potentially very interesting to elucidate newly-found knowledge related to roles of various elements in bone.
However, the potential is, for me personally, marred by several points:
1. The review should outline more clearly why elements were chosen - define the aspect or angle from which this review will cover them. This is especially important for elements which have been extensively reviewed in the past.
The approach could otherwise seem too general and perhaps not very novel.
I would also suggest to modify the title in this regard and to also use full element names instead of periodic table symbols as to aid the visibility of the paper in search engines.
2. The flow of the manuscript feels somewhat disjointed as an overview for a particular element starts with general information, then heads into newer research results, and then sometimes returns into general statement(s) (which would be more welcome in the introductory part for the element anyway).
I would advise reorganizing the order of information presentation as to try and keep the previously established data and knowledge to the introductory part for an element, as well as to try and "string" the information together in a more fluid way.
Also, as mentioned above, I would strongly advise keeping emphasis on discussion of the referenced newer studies and reducing relatively known facts to a necessary minimum.
3. Some statements which should be argumented with study references are not. One bigger example is paragraph from line 138 to 145.
I would advise going over through the content and making sure to correct these occurences.
4. I am aware that a lot of work went into the tables, but I am not sure how much the tables in this form contribute to the review itself. I am under the impression that the key information could be presented more succinctly and clearly in a comparatively smaller text paragraph.
Thus, I would advise to try and reduce size of all tables and summarize them as much as possible - ideally into one joint table if possible. I would have personally more liked to have read the authors discuss the sometimes big differences between the noted studies, as well as the notes in the "additional information" columns.
5. There are at least of couple of occurences I have spotted where parts of text or even paragraphs are almost same as in the referenced study.
One example from the manuscript: "Owing to its cumulative effect in the body, the concentration of lead in the tissues changes throughout a person's life. In adults 90–95% of total body lead is stored in trabecular and cortical bone, while in children 70–95% accumulates in trabecular bone due to its high turnover rate [152]. During menopause, as well as pregnancy and lactation, a rise in blood lead levels stimulates its accumulation in other organs [153,144]. For that reason, the concentration of lead in the blood and soft tissue organs may change throughout life depending on the metabolic activity of bone tissue."
Compare it to Rodríguez J, Mandalunis PM. A Review of Metal Exposure and Its Effects on Bone Health. J Toxicol. 2018;2018:4854152:
"Due to its accumulative effect in the body, tissue concentration of lead varies throughout an individual's life. In adults, 90-95% of total body lead is stored in trabecular and cortical bone, whereas, in children, 70 to 95% accumulates in trabecular bone as a result of its high turnover rate [77]. During pregnancy [78], lactation, and menopause, the increase in blood levels of lead stimulates lead accumulation in other body organs [79]. Therefore, the concentration of lead in blood and soft tissue organs can vary throughout life according to the metabolic activity of bone tissue."
Even though the publications are referenced by this work, I would strongly advise rewriting all such occurences.
This issue may also contribute to the feeling that it is apparent to a reader that different authors wrote different paragraphs, as well contributes to the "disjointed" feeling I noted under point nr. 2.
6. Under "Conclusion" - please correct "...lead are crucial for bone tissue homeostasis".
In conclusion, I would commend the amount of effort and the amount and quality of adequate references with novel data, but would recommend a more thorough revision of the whole text in line with above comments.
Author Response
Dear Editor,
Subject:
Thank you for your email dated 14 March 2021 enclosing the reviewers’ comments.
We have carefully reviewed the comments and have revised the manuscript accordingly. Our responses are given in a point-by-point manner below. Changes to the manuscript are shown in red.
We hope the revised version is now suitable for publication and look forward to hearing from you in due course.
Sincerely,
Dr Karolina Kot
Response to Reviewer 2:
Thank you for your review of our paper. We have answered each of your point below.
The topic of this review is potentially very interesting to elucidate newly-found knowledge related to roles of various elements in bone. Thank you
However, the potential is, for me personally, marred by several points:
1. The review should outline more clearly why elements were chosen - define the aspect or angle from which this review will cover them. This is especially important for elements which have been extensively reviewed in the past. The approach could otherwise seem too general and perhaps not very novel.
I would also suggest to modify the title in this regard and to also use full element names instead of periodic table symbols as to aid the visibility of the paper in search engines. The choice of these elements was guided by the enormous importance of calcium, magnesium, phosphorus, fluorine, and lead on bone tissue. Apart from the mechanism and the influence of these elements on bone tissue, which have been presented in other publications, our goal was to focus on their specific concentration. Yes, so far some researchers have tried to take up a similar topic, however, the lack of specific reference values of elements in the bone tissue makes the interpretation of these results very difficult. We used names instead of periodc table symbols in the title.
- The flow of the manuscript feels somewhat disjointed as an overview for a particular element starts with general information, then heads into newer research results, and then sometimes returns into general statement(s) (which would be more welcome in the introductory part for the element anyway).
I would advise reorganizing the order of information presentation as to try and keep the previously established data and knowledge to the introductory part for an element, as well as to try and "string" the information together in a more fluid way. Also, as mentioned above, I would strongly advise keeping emphasis on discussion of the referenced newer studies and reducing relatively known facts to a necessary minimum. Each subsection has been revised in line with the comments. - Some statements which should be argumented with study references are not. One bigger example is paragraph from line 138 to 145. I would advise going over through the content and making sure to correct these occurences. We added citation.
- I am aware that a lot of work went into the tables, but I am not sure how much the tables in this form contribute to the review itself. I am under the impression that the key information could be presented more succinctly and clearly in a comparatively smaller text paragraph. Thus, I would advise to try and reduce size of all tables and summarize them as much as possible - ideally into one joint table if possible. I would have personally more liked to have read the authors discuss the sometimes big differences between the noted studies, as well as the notes in the "additional information" columns. Due to a large number of informations in the tables, we decided to leave it in the manuscript. Accorfing to Reviewer 1, we added new information to the Tables. Also, some of the additional information from the tables has been added to the text to make it easier to read.
- There are at least of couple of occurences I have spotted where parts of text or even paragraphs are almost same as in the referenced study.
One example from the manuscript: "Owing to its cumulative effect in the body, the concentration of lead in the tissues changes throughout a person's life. In adults 90–95% of total body lead is stored in trabecular and cortical bone, while in children 70–95% accumulates in trabecular bone due to its high turnover rate [152]. During menopause, as well as pregnancy and lactation, a rise in blood lead levels stimulates its accumulation in other organs [153,144]. For that reason, the concentration of lead in the blood and soft tissue organs may change throughout life depending on the metabolic activity of bone tissue."
Compare it to Rodríguez J, Mandalunis PM. A Review of Metal Exposure and Its Effects on Bone Health. J Toxicol. 2018;2018:4854152: "Due to its accumulative effect in the body, tissue concentration of lead varies throughout an individual's life. In adults, 90-95% of total body lead is stored in trabecular and cortical bone, whereas, in children, 70 to 95% accumulates in trabecular bone as a result of its high turnover rate [77]. During pregnancy [78], lactation, and menopause, the increase in blood levels of lead stimulates lead accumulation in other body organs [79]. Therefore, the concentration of lead in blood and soft tissue organs can vary throughout life according to the metabolic activity of bone tissue."
Even though the publications are referenced by this work, I would strongly advise rewriting all such occurences.
This issue may also contribute to the feeling that it is apparent to a reader that different authors wrote different paragraphs, as well contributes to the "disjointed" feeling I noted under point nr. 2. We rewrite this part of manuscript.
- Under "Conclusion" - please correct "...lead are crucial for bone tissue homeostasis". We changed it.

Round 2
Reviewer 2 Report
I thank the authors for revising the manuscript according to most of my comments.
I fully agree that the greatest strength and addition of this work is definitely "specific reference values of elements in the bone tissue", which the authors state in their reponse as the main goal and focus of their review.
Thus, I still feel it is a missed opportunity not to rewrite at least the title and introduction in this regard - to point out their focus and greatest novelty of this work. It would be much more visible and gather more citations in my opinion.
However, this is now a question of style and approach, I will not insist further on what is now in domain of editorial decision.
Small final note - line 19 "concentration of each of these elements un the
various bone tissue." needs to be corrected.